# Evaluation of Improvements in the Separation of Monolayer and Multilayer Films via Measurements in Transflection and Application of Machine Learning Approaches

**DOI:** 10.3390/polym14193926

**Published:** 2022-09-20

**Authors:** Gerald Koinig, Nikolai Kuhn, Chiara Barretta, Karl Friedrich, Daniel Vollprecht

**Affiliations:** 1Chair of Waste Processing Technology and Waste Management, Department of Environmental and Energy Process Engineering, Montanunigersitaet Leoben, Franz Josef Straße 18, 8700 Leoben, Austria; 2Polymer Competence Center Leoben GmbH, Roseggerstraße 12, 8700 Leoben, Austria; 3Chair of Resource and Chemical Engineering, Augsburg University, Am Technologiezentrum 8, 86159 Augsburg, Germany

**Keywords:** 2D plastic packaging, near-infrared spectroscopy, sensor-based sorting, transflection, monolayer, multilayer films, machine learning, small film recycling

## Abstract

Small plastic packaging films make up a quarter of all packaging waste generated annually in Austria. As many plastic packaging films are multilayered to give barrier properties and strength, this fraction is considered hardly recyclable and recovered thermally. Besides, they can not be separated from recyclable monolayer films using near-infrared spectroscopy in material recovery facilities. In this paper, an experimental sensor-based sorting setup is used to demonstrate the effect of adapting a near-infrared sorting rig to enable measurement in transflection. This adaptation effectively circumvents problems caused by low material thickness and improves the sorting success when separating monolayer and multilayer film materials. Additionally, machine learning approaches are discussed to separate monolayer and multilayer materials without requiring the near-infrared sorter to explicitly learn the material fingerprint of each possible combination of layered materials. Last, a fast Fourier transform is shown to reduce destructive interference overlaying the spectral information. Through this, it is possible to automatically find the Fourier component at which to place the filter to regain the most spectral information possible.

## 1. Introduction

Currently, around 300,000 t of plastic waste are annually produced in Austria, of which 32% are recycled mechanically [1]. Small films with an area below 1.5 m^2^ account for 69,000 t, of which 10,260 t, or 14%, are multilayer films with at least two polymers [2]. These films are separated during the beneficiation of the waste and are almost exclusively used as alternative fuel sources, incinerated or downcycled into low-value products [3]. The substantial potential is latent in the recycling of packaging films since neither co-incineration nor other thermal recovery adds to the recycling quota [2]. According to the amended EU Waste Framework Directive, municipal solid waste recycling must reach 60% by 2030 [4]. Additionally, the new EU guidelines require a recycling rate of packaging waste of 50% in 2025, with a further increase to 55% in 2030 [4]. This quota can only be reached through a mix of measures such as higher collection rates, design for recycling, and improving existing and new sorting techniques. Besides, a recycling system capable of economically handling a feedstock which accounts for 17 wt.% of all plastic packaging materials produced, requires additional research. [3].

The reason for the widespread use of multilayer packaging lies in its convenience for producers, retailers and consumers: The plethora of functions such as UV protection, handleability, printability, limited gas permeability, and attractive haptics require only a minimum of packaging material.

In turn, these inherent properties, the thin layer thickness and the combination of different polymers impede the separation process. In most sorting plants, near-infrared (NIR) sorters are used for plastics separation. This technology is based on the interaction of NIR radiation with the molecular structure of solid materials resulting in distinctive spectral fingerprints for each polymer type [5]. Thin-film packaging inhibits the separation by NIR sorting because only a limited amount of radiation is reflected [6]. This lack of spectral information limits the sorter’s ability to generate a useable spectrum because the low thickness of the material allows a large amount of radiation to be transmitted [7]. Additionally, the thin layered construction of these packaging films introduces disturbances in the spectral fingerprint. Due to destructive interferences, sine wave pattern noise may overlay the spectra, masking its information and thus disfiguring an otherwise applicable spectrum [8]. Fast Fourier transformation (FFT), which is also used in laboratory-based infrared spectroscopy, can reduce these overlaying interferences. Though finding the correct cut-off point has proven to be both time-consuming and tedious if carried out manually [8].

The resulting lack of spectral information can lead to misclassified particles, which in turn could contaminate an otherwise clean feedstock. This contamination impedes the recyclability of the recyclate by altering its mechanical properties. This alteration may result in the need for additional compatibilisers and other additives for the intended recycling process. [9].

As NIR-based sorters are most widely used in sorting plants, but their potential has not yet been fully exploited, the aim of this research was to improve their material detection. Additionally, the decoupling of the material properties from the mechanical separation enables not only a change of the hardware configuration, in this case the measurement in reflectance mode, but also the software of the evaluation unit.

Given that a simple adaptation of existing sorter may improve their capability to separate thin, flexible packaging material, substantial increases in recycling quota with a limited investment are feasible. Preliminary studies have shown the possibility of separating monolayer from multilayer materials on a laboratory scale using a NIR-active background in an experimental setup [7]. Further examinations of these findings on an industrial scale NIR Sorter have proven to increase the spectral quality of flexible packaging films by implementing a metal reflector to the sorting geometry [2]. Implementing a NIR inactive metal sheet as a reflector enabled the sorter to measure in transflectance rather than the usual reflectance mode [2]. 

Apart from the low material thickness, another prevalent advantage of multilayer films has proven problematic during separation and recycling: the continuously changing types of polymer types, thickness and sequence to ensure the best product protection. Hence, the resulting combination possibilities further complicate the creation of separation models.

Whenever completing a complex task without explicitly programming every conceivable variation of this task, machine learning becomes the tool of choice. The application of machine learning methods in NIR spectroscopy has been successfully implemented in various fields. It has been used to assess the quality of beer from given features [10], the rapid assessment of water pollution [7] or the prediction of soil total nitrogen, organic carbon and moisture [11].

This paper investigates the effects of adding a reflective chute material to a state-of-the-art near-infrared sorting unit. This modification allows 2D plastic packaging material consisting of single and multi-layer films to be more effectively detected via transflection and subsequently separated. In addition, an automatic method for applying the FFT to spectra obtained in this transflection configuration and affected by interference is examined. This method is an alternative to manually determining the correct cut-off point in the Fourier deconstruction of the spectra. Based on these improved spectra, a principal component analysis is performed to evaluate whether there are predominant spectral differences between spectra of mono- and multilayer materials. This characteristic difference can be used to train machine learning algorithms to separate the two fractions. 

Machine learning algorithms are then evaluated based on their prediction performance and calculation speed. These metrics result in a hierarchy gauging their capability to produce correct predictions in a reasonable time. This examination is necessary to gauge whether this approach is feasible for inline applications, categorising spectra generated in an industrial environment. Finally, an integrated method is discussed, using improved spectral recognition with mechanically adapted NIR sorter, improved spectra rid of sine wave interferences and separated into mono- and multilayer materials via supervised machine learning classification algorithms.

The presented information creates a stepping stone for integrating recyclable resources to increase the effectiveness of mechanical recycling. This increase in effectiveness further creates a value-adding raw material source for multilayer recycling processes currently in development, thus improving the circular economy of polymers [12,13].

## 2. Materials

All experiments were executed with material obtained directly from the input of an Austrian material recovery facility. This waste was collected under the Austrian lightweight packaging collection scheme. Under this scheme, lightweight packaging made of polymers, aluminium or beverage carton is collected. For plastic packaging, the collection includes both 3D and 2D material. From this material, the film specimen for this research paper were sampled.

### 2.1. Film Specimens

A total of 103 specimens of post-consumer waste were taken directly from a sorting plant’s input fraction in Austria. The input fraction is delivered in yellow bags, and these bags were collected and the lightweight packaging therein was used for further evaluations. The samples were neither cleaned, smoothed or otherwise exposed to preparatory conditioning before the sorting trials were conducted. The samples’ dimensions ranged from 10 mm × 10 mm to 210 mm × 297 mm and included printed and transparent samples. Figure 1 shows the small film fraction for reference.

An examination with Fourier transform infrared (FTIR) spectroscopy yielded the material composition of the experimental samples. The spectrometer used was a Spectrum Two FTIR spectrometer (Perkin Elmer, Waltham, MA, USA) equipped with a Zn/Se crystal with a diamond tip. The spectrometer measures in the range of 650 cm^−1^ to 4000 cm^−1^ and has a spectral resolution of 4 cm^−1^. 

#### 2.1.1. Classification with FTIR Spectroscopy

The exact measurement method is explained in greater detail in a paper published by Koinig et al. in 2022, which examined the composition of Austrian lightweight packaging waste using FTIR measurements. The method is therefore described in short in the following.

Fourier-transform infrared spectroscopy (FTIR) in attenuated reflectance (ATR) mode was used to classify the film specimen into their respective material classes. 

Samples on which the results differ for the front and back are defined as multilayer films, while samples with identical results for the front and the back are defined as monolayer films. However, the FTIR-ATR characterisation method is limited to identifying the polymeric material on the sample’s surface and penetrates only a few micrometres of the sample thickness. In case of uncertainties in assigning a sample to the mono- or multilayer category, additional FTIR measurements were performed in transmission mode to investigate the material composition over the entire sample thickness to ensure reliable results.

According to the FTIR spectral analysis, the specimens were categorised into different groups of mono- and multilayer materials. The materials represented by the selection of samples are represented in Table 1.

#### 2.1.2. Experimental Sensor-Based Sorting Setup

The trials were conducted with an experimental sensor-based sorting (SBS) setup. The NIR sensor, an EVK-Helios-G2-NIR1, was used for the trials. This sensor detects the reflected NIR radiation emitted by a halogen lamp on a sample. The emitted radiation is reflected, absorbed, or transmitted depending on the specimen and interacts with near-surface molecules [14]. The spectral resolution of the sensor is 3.18 nm with a frame rate of 476 Hz and an exposure time of 1800 µs. Each spatial pixel is 1.60 mm wide, owing to the geometrical setup of the testing rig. The waveband evaluated during the trials was 991 nm to 1677 nm, split into 220 discrete measuring points. After detection, the radiation is analysed with EVK SQALAR to classify the respective spectra.

The function principle of the sorting rig is depicted in Figure 2. 

#### 2.1.3. Reflectors

The sorting experiments, which were the basis for the data evaluated in this paper, were conducted with two reflective chutes made of aluminium and copper. These adaptations had to be made to the existing sorting setup to allow for measurement in transflection. Two variants of the reflective chute were manufactured by laser cutting the metal plates. The specific shape of the reflector was chosen so as not to cover the illumination of the sorter, which is necessary to detect objects for ejection. Copper and aluminium were used as reflective materials because they are highly promising due to their high reflectivity of NIR radiation [15].

## 3. Methods

The described experimental sensor-based sorting setup was used to classify the 2D materials during the trials. This chapter explains the preparations to complete the sorting model generation and separation of materials. Further, the measurements in transflection mode are explained. Finally, the methods used in creating the machine learning approaches and the spectra improvement methods are explained.

### 3.1. Measurements in Transflection Mode

One of the defining characteristics of NIR sorting is the interaction of material and NIR radiation. During this interaction, the incident radiation energy is partially converted into kinetic energy of molecular vibrations, while other parts of the radiation’s intensity are transmitted and reflected [16]. Only sufficient interaction between the molecules of the specimen and the incident NIR radiation creates useable NIR spectra for classification. Material with insufficient thickness causes most of the incident radiation to be lost to transmission. Additionally, the minuscule amount of reflected radiation has not interacted sufficiently with the material to cause alterations in the spectra. Preliminary studies have shown that the minor signal alterations caused by the low material thickness in reflectance mode can be alleviated by adapting the experimental sensor-based sorting setup for measurements in transflectance mode [7,17].

Placing a reflective background plate onto the chute allows measurements to be taken in transflection mode. This way of measuring thin films alleviates the problem caused by the low thickness of the material. The radiation is reflected after its first pass through the specimen. This approach enhances the interaction of radiation and material because of the lengthened path the radiation takes through the material: First, the incident NIR beam enters the sample and a small proportion of its intensity is immediately reflected. However, a significant proportion is transmitted through the specimen and consequently reflected by the reflective material placed behind the sample. Hence, it passes again through the material and can be detected through the NIR sensor. This additional pass through the material increases the spectral quality and enables the creation of a sorting model to classify film materials.

Through this process, the variability of the spectra is decreased. This variability is defined as the absolute difference between pixels of the same specimen [2]. 

Only if the pixels of a given specimen exhibit similar spectra, a specimen be classified correctly. Figure 3 compares the spectra of a PE film measured in transflectance mode (left) and the standard reflectance mode (right). The depiction shows the mean spectra of ten pixels, normalised via the “zScore” method and smoothed by Gaussian smoothing with a 10-point floating smoothing window. It can be seen that the characteristic PE peak at 1150 nm becomes more pronounced when measured in transflection.

### 3.2. Preparations for Sorting Trials

The trials were conducted with teaching and testing fractions. The specimens were separated into a teaching set to create the model containing 80% of the materials and a separate testing set to check the model prior to the sorting trials containing 20% of the specimen. A train test split of 80:20 is one of the most effective ways to train models [18]. The train set consisting of known composition mono- and multilayer materials was used to create a sorting model. The second class was the test set consisting of monolayer and multilayer materials not used for teaching the sorting model. With the teaching and test sets created, the reflective background was installed, and the sorting model, which is necessary to classify and eject the multilayer materials, was created.

#### Model Creation Using EVK SQALAR

The sorting model for separating the individual materials was created using EVK SQALAR.

A sorting model for NIR sorting defines the criteria for which the experimental sensor-based sorting setup sorts fractions based on reference spectra. These spectra are taken from known composition materials, and these benchmark spectra are compared to the unknown materials’ spectral information during the sorting trials. If an unknown pixel’s spectrum shows sufficient similarities to a reference, it is classified as this material class. 

Apart from the reference spectra, the sorting model defines the pre-processing and spectral processing methods applied to the spectral information. Here, the upper and lower limits of spectral intensity in which viable pixels for evaluation lie, are considered. Concessions were made to create a sorting model that can use reflective backgrounds. Firstly, the white calibration with the reflective background was completed, allowing the existing white calibration algorithm to adapt to the increased intensity of reflected radiation due to the adapted chute material. Secondly, the illumination intensity had to be lowered to prevent overexposed pixels. This was performed despite the results of previous research stating that increased illumination intensity improves the spectral quality [17].

Table 2 shows the pre-processing and spectral processing methods used in preparing the spectral information for classification. These methods were described in the literature as ideal for separating post-consumer waste as they enhance the subtle differences in each spectrum, facilitating the differentiation between similar spectra, for example, between PE or PP monolayer and PE–PP multilayer [19,20].

This procedure for creating a sorting model was undertaken with the standard configuration for measurements in reflectance while the aluminium and the copper reflectors were used for measurements in transflectance. This approach yielded an individual sorting model for reflective surfaces and the non-reflective original chute. 

### 3.3. Sorting Trials

The sorting trials were performed with every specimen in the test set. Each attempt was repeated five times to eliminate random factors, such as the trajectory of the film specimen. The sorter was set to eject multilayer materials. 

A particle was considered to be classified correctly when the high-pressure nozzles were activated and the particle was ejected. Through this approach the number of correctly separated specimens for the respective configuration.

### 3.4. Principal Component Analysis to Determine the Possibility of the Application of Machine Learning Approaches

Even with increased fidelity to the material’s spectral fingerprint in the available spectra, the overabundance of available multilayer material combinations poses a problem in creating a sorting model. It is infeasible to implement a sorting routine with spectral information to correctly recognise all available multilayer material to differentiate it from monolayer material, and neither is it feasible to include all existing monolayer materials in the sorting model. Therefore, it is necessary to adopt a sorting mechanism that achieves the task of detecting multilayer materials without explicitly implementing a vast number of multilayer spectra. For this, a supervised learning approach was chosen. In order to achieve this, common identifying characteristics of multilayer materials must be present. If they influence the spectra enough to enable classification, the existence of these characteristics would enable the separation of multilayer materials without the need to gather the spectra of each material. A principal component analysis (PCA) was applied to the 17,569 spectra recorded from the multilayer and monolayer specimens. The PCA was used to reduce the 220-dimensional spectral information into principal components to analyse if sufficient differences are present in the data to explain the variance of the data set with principal components.

Since the PCA indicated differences between multilayer and monolayer spectra, a comparison of the average of the multilayer material and monolayer material spectra was conducted. This comparison was used to evaluate the spectral range in which the two classes differ most. This comparison was made by taking the mean of all multilayer and monolayer spectra used in this trial. The two resulting spectra were compared by taking the two-norm of the distance of each spectral point of the monolayer spectra from its corresponding spectral point of the multilayer spectra. This yielded in the wavelengths at which monolayer spectra and multilayer spectra differ substantially from each other. 

### 3.5. Evaluation of Machine Learning Approaches to Classify Spectral Data

With the information gathered via the PCA, an array of machine learning approaches was applied to the spectral information gathered from the thin film specimen. First, the 17,569 spectra gathered from the specimen were randomly separated into a training and a test set. This is required to enable holdout validation to train the machine learning approaches. The test set contained 20% (3513) of the spectra, while the training set consisted of the remaining 80% (14,056) of spectra, again utilising the recommended 80/20 split.

Cross-validation allows training and testing on a given number of data splits and thus permits an estimate of how well a given model will perform on unseen data. Holdout validation depends on splitting the data set according to the given ratio between the training and test set. Even with cross-fold validation potentially increasing the prediction success by 0.1–3%, the time trade-off on large data sets is substantial [21]. The machine learning approach was repeated using cross-fold validation with five cross-validation folds. One of the selected groups is used as a test set, while the other is used as a training set. After grouping, the model is trained on the training set and tested and scored using the test set. This process is repeated until all sets have been used as the test set. The holdout validation was chosen after preliminary tests resulted in a high prediction success when using holdout validation while requiring less training time.

Each NIR spectrum consists of 220 spectral data points. Every spectral point contains the radiation intensity detected by the NIR sensor and is a feature used for predicting the material class in this context. The first derivative of every spectrum was taken to enhance differences inherent in the spectral data, and no further feature engineering, e.g., feature selection, was performed. Thus, the machine learning approach initially used all 220 spectral points equally spaced over the NIR spectral region of 930–1700 nm. After these preliminary trials, a PCA was conducted, reducing the number of features from 220 to 3. These three features explained ~80% of the variance in the model. This approach tripled the number of observations per second the models were capable of and increased the prediction accuracy in one case. 

All necessary computations were conducted running MATLAB by The MathWorks (Natick, MA, USA) Version 9.10.0.1710957 (R2021a) Update 4 on a Windows 10 computer equipped with an Intel^®^ UHD Graphics 630 and an Intel ^®^ Core ™ i5-9400H CPU clocked at 2.50 GHz.

#### 3.5.1. Used Machine Learning Algorithms

Supervised learning approaches were used to differentiate between mono- and multilayer materials. The selection process for the correct algorithm yielded several different machine learning approaches to be tested. Since the problem at hand is a clustering problem with three possible clusters, the following algorithms were chosen and evaluated for their performance.

##### Decision Tree

Decision Trees are known as Classification and Regression Algorithms since they can perform classification and regression. Decision trees follow along their edges or branches and decide at the nodes which branch to follow to label a new input. A condition is queried at every node to decide which branch to follow [22]. When categorising whether a given material is a multilayer film or not, the prime features to be evaluated are the intensity of the given pixel at a specific wavelength. Figure 4 shows an example of a simple decision tree.

##### k-Nearest Neighbour

The k-nearest neighbour (kNN) algorithm works by analysing the distance between a new data point and its k-nearest neighbours. The user determines the number of neighbours evaluated, k, influencing the algorithm’s outcome. The new data point is then assigned the label of the majority of its neighbours. The Euclidian distance between neighbouring data points is used as a decision criterion. [23].

Figure 5 shows an example; if k = 5 and 3 neighboring points are classified as multilayer while two are classified as a monolayer, the new data point will be labeled multilayer. In this example, the dimensionality has been reduced from 220 to 2 by a prior PCA. This reduction in dimensions is usually made in preparation of a kNN approach to avoid the effects of the curse of dimensionality, which plagues many machine learning algorithms [24]. In kNN, the Euclidian distance becomes useless as a metric in higher dimensions since all vectors are equidistant to the search query vector. 

##### k-Means

The k-means algorithm is well suited to classification problems. It works by defining a number, k, of clusters. Then a set of centres for those clusters is randomly selected. All data points are then labeled according to their distance to these clusters. After this clustering, the new centres of those clusters are calculated, and the algorithm begins anew, again clustering the data around the new cluster centres. With every iteration of the algorithm, the change of the centres becomes smaller. The procedure is repeated until a threshold number of iterations is reached. The classification is then complete, and the model can be used to classify new data according to the k-clusters. The success of this approach dramatically depends on the selection of the initial centres. It is therefore advisable to create various k-means models with different starting parameters. Apart from relying on the starting conditions, the k-means approach’s low computational and memory requirements are its advantages. Figure 6 shows a completed clustering using the k-clusters approach.

##### Support Vector Machine

Support vector machines (SVM) separate the given data set by a hyperplane that maximises the empty area between different data sets. This area is called the margin. The solution offering the maximum margin separating the given data sets is considered the optimum and chosen to classify new data. These separating lines, or hyperplanes, are generated by support vectors, thus the name. A sample showing the classification process of an SVM is shown in Figure 7. These hyperplanes can be linear and not linear, rendering the SVM able to classify most data sets of natural features where a linear separation is impossible [25].

##### Neural Net

The application of neural networks for classification differs from traditional machine learning algorithms. A classification task requires the input of labeled data, and this supervised learning approach can be used to classify all data that humans can label. Neural networks are commonly applied to text classification, fraud detection, voice identification, or video analysis. A shallow neural network (SNN) with one connected layer has been applied to the input. The input consisted of the first derivative of the spectra contained in the spectral image. The classification yielded three classes for the evaluated pixels: multilayer, monolayer and background.

#### 3.5.2. Feature Engineering

Before classification, the raw spectral data was normalised using the “zScore” method, which ensures a mean value of zero and a standard deviation of one. The spectra were smoothed using a Gaussian smoothing algorithm with a ten-element sliding mean window. Additionally, the first derivative of the spectral data has been taken to make the differences inherent in the spectral data more prominent.

### 3.6. Use of Fast Fourier Transformation to Improve Spectra

Fast Fourier transformation (FFT) was applied to improve spectral quality. This approach enabled overlying sine wave-like spectral abnormalities to be reduced. This reduction in overlying sine wave-like spectral abnormalities made the analysis of previously obscured spectral information possible. 

The fast Fourier transformation algorithm of MATLAB is used to achieve the original spectrum’s discrete Fourier transformation (DFT). The DFT of a signal decomposes the original spectrum into a series of harmonic sinewave parts and represents a frequency spectrum. Figure 8 shows a representation of a generic noisy signal. Here it can be seen that any signal composes itself of a series of overlying frequencies. The underlying signal is overlaid with noise, making it difficult to determine the original signal. The noise could be eliminated by manipulating the signal in this representation, making the signal clearer.

By manipulating the representation of the original spectrum, unwanted noise, for example, the aforementioned sine wave abnormalities, can be omitted in the inverse Discrete Fourier transformation (iDFT). The iDFT is used to recreate the signal. To generate a usable spectrum, the placement of this filter has to be evaluated, and the resulting spectrum has to be compared to a suitable reference spectrum. This computation takes the two-norm of the difference between the new spectrum and the reference spectrum. An algorithm evaluates the resulting spectrum concerning the reference spectrum and places the filter in the position that yields the optimum spectrum, which facilitates this evaluation. This way, manual experimentation of filter placement, which previously took considerable time, can be automated [8]. The deviation from the reference spectrum is plotted over the corresponding filter position for visual inspection to evaluate the correct positioning. The result of this process is shown in Figure 9, which depicts the evaluated placement point for the low pass filter and the resulting deviation. The point in the search with the lowest deviation is marked. This placement point was then used for further processing.

The original spectrum is represented in 220 Fourier coefficients. These coefficients correspond with the camera’s spectral resolution with which the spectrum was recorded. Figure 10 shows the representation of the spectrum after the FFT was applied. Further, the location of the deep pass filter is visualised.

#### Summary of Applied Methods

In summary, three methods were used to solve problems in sorting films. Firstly, the spectra quality was insufficient for separating the material. This issue was remedied by applying measurement in transflection. The second problem was that after the inclusion of reflective backgrounds for measurement in transflection, sine wave-like disturbances were still occurring in the spectra. These in turn were reduced with FFT. Because finding the correct cut-off point for the low pass filter by hand is time-consuming, an algorithm is used which finds the cut-off point that results in the best spectra after reconstruction.

The third problem was the abundance of material compositions in multilayer films, which impeded the creation of a sorting model that recognises multilayer materials. PCA and the comparison of multilayer film and monolayer film spectra evaluated the viability of applying machine learning methods to solving this problem. With these methods, characteristic differences in the spectra were found, which promised a successful application of machine learning methods. These methods were used to classify film spectra into two groups and were compared to each other’s prediction accuracy and computation speed to find the best machine learning method suited for the task. Figure 11 shwos a summary of encountered problems when sorting films and the applied solutions.

## 4. Results

All results are assessed based on the number of correct ejections. The first sorting trials were conducted without adaptations to the sorting rig. These results are used as reference values to compare the effect of introducing a reflective background on sorting multilayer and monolayer films.

### 4.1. Detection Rate without Reflector (Glass Chute)

In summary, 46% of all materials were correctly sorted using no reflective surface as a background for classification. The lack of useable spectral information explains this low sorting success. Given the lack of spectral information to base the classification, creating the sorting model proved difficult. 

Figure 12 depicts the detection rate for all materials using no reflective background.

### 4.2. Detection Success with Aluminium Reflector

The second trial was conducted with the use of an aluminium reflector. 

Due to the optical properties of aluminium, it reflects near-infrared radiation and permits measurements in transflection mode. Since less radiation is lost to transmission, more pixels contain useable spectral information for classification. This effect permits the detection of the 2D materials, independent of their thickness and coloring. Optically transparent materials are detectable and, therefore, sortable with a reflective surface. This improved sortability is shown in Figure 12, which depicts the detection rate for all materials using an aluminium reflector and compares it to the initial results without a reflector. After the trial, 74% of all multilayer materials were ejected correctly, which is 61% more compared to the measurement in reflection. 

The aluminium reflector showed great promise as a reflective surface, although its tendency to accrue an oxide layer that diminishes its reflective capabilities needs to be considered.

### 4.3. Detection Success with Copper Reflector

The third trial was conducted using a copper plate as a reflective surface. The high reflectivity of copper facilitated the model creation. Due to the high reflectivity, the number of useable spectra for model creation was increased. The copper’s reflectivity enabled a sorting model that successfully distinguished the majority of mono and multilayer materials used in the trials. Figure 12 shows a comparison between all three setups. 

### 4.4. Comparison of the Detection Experiments

In addition to its inherent higher reflectivity in the NIR spectrum compared to aluminium, copper does not tend to create an oxide layer, and this property may make it more viable as a reflector despite its higher cost relative to aluminium. Figure 12 shows the overall detection rate for all materials on all reflectors as a comparison and the total percentage of detected objects. The formation of verdigris was not encountered during the trials but will most likely pose an issue when using the reflective surface in an industrial setting and must be included in planning.

### 4.5. Evaluation of Differences in Ejection Rate between Polymer Types

This chapter explains the differences in detection and subsequent ejection between the polymer types. The spectral differences causing this lack of uniformity in ejection are explored. For this purpose, spectra were taken in each measurement mode (RAW), standard measurement without reflector, transflectance with aluminium reflector (AL-TR), and transflectance with copper reflector (CU-TR), are shown and compared with each other. In addition to the mean spectrum of the specimen, the variability of the spectrum is shown. The lower this spectra variability is, the easier the specimen can be assigned to one material group. In Table 3, the results of the trials are presented in tabular form for ease of reference in the comparison.

#### 4.5.1. PE

It can be seen that 40% of PE films were falsely classified as multilayer materials and ejected when using no reflector. Figure 13 shows the spectra of a PE specimen used in the trials and shows that the variability of the spectrum taken without a reflector is comparably high. Especially in the area of 1200 nm, the second characteristic PE section is absent and diffuse. The spectra recorded using the copper reflector show very sharp characteristic sections at 1200 nm and 1400 nm with little variability. The spectra are shown in Figure 13.

#### 4.5.2. PP

PP was recognised much better than other plastics in the trials. Without a reflector, 83% of the specimen were correctly sorted. Implementing an aluminium reflector raised this to 90% while implementing a copper reflector reduced the result to 77%. The answer to this abnormal behavior can be found in the spectral analysis. Examining the spectra taken in AL-TR, it can be seen that three characteristic peaks are present, namely at 1300 nm, 1400 nm and 1550 nm. In RAW, only one of those characteristics is present. In CU-TR, two of these three sections are present and can be used for classification, with the dip at 1550 nm absent. Irrespective of the used reflector, the quality of the PP spectra is more susceptible to the thickness of the specimen. PP specimens exhibit sine wave-like noise disturbances of the spectra at a higher thickness than other polyolefins such as PE [8,17]. The spectra are shown in Figure 14.

#### 4.5.3. PE/PA

PE/PA showed a slight improvement in spectral quality. The characteristic regions at 1200 nm and 1400 nm are present without or with the reflector. The dismal ejection rate without the reflector was due to misclassification as a monolayer since the PE is dominant in the PE/PA spectrum. It can be seen that an aluminium reflector alters the spectrum in the region of 1200 nm by extending the peak in comparison to measurements without a reflector or with a copper reflector. The spectra are shown in Figure 15.

#### 4.5.4. PE/PET

PE/PET had the worst detection rate, with an average of 33% of all specimens correctly ejected in all trials. The cause is that PE makes up the central part of PE/PET composites. Since the intensity of any spectral component is proportional to the material’s thickness, what little spectral information is detected resembles PE [26]. This dominance of the PE spectrum leads to the misclassification of the multilayer material as PE monolayer and, subsequently, the low sorting accuracy. These spectra can be seen in Figure 16. In the spectra recorded in RAW, the characteristic PET dip at 1500 nm is blurred by the variance. Measured with AL-TR, the characteristic PE peak is blurred, and the PET dip is diminished while the dip at 1400 is present. In CU-TR, all characteristic features of the PE PET multilayer are sharp and easily distinguishable, leading to the correct results.

#### 4.5.5. PP/PET

PP/PET was one of the films with the lowest ejection rate. It can be seen in Figure 17 that the characteristic PET peak at 1650 nm only starts to appear when a copper reflector is used. The spectra recorded without a reflector exhibit no spectral information and are unsuitable for classification. The inclusion of an aluminium reflector improves the spectra to a limited extent. More pronounced improvements are reached after a copper reflector was installed. After this installation, the characteristic peaks of PP and PET become apparent, reducing the risk of misclassifying the films as PP. The spectra are shown in Figure 17. 

#### 4.5.6. PE/PP

PE/PP multilayer specimens were sorted out without a reflector 40% for the time, and the introduction of AL-TR raised this to 72%, and CU-TR further increased this result to 76%. PE PP multilayer is especially susceptible to high variability in the spectrum since it is a composition of two materials exhibiting similar NIR spectra. All peaks overlap the PE and PP spectra and are present in CU-TR; thus, the material can correctly be classified as a multilayer film. The spectra are shown in Figure 18.

#### 4.5.7. PP/PA

PP/PA was comparatively well separable without a reflector. The spectra taken in RAW exhibit the material’s characteristic peaks and minimal variability. The sharpness of the characteristic peaks and the variability of the spectra were further improved when measuring in AL-TR or CU-TR, mirrored by the improved results in the sorting trials. The spectra are shown in Figure 19.

### 4.6. Application of Machine Learning Algorithms to Classify Film Spectra into Multilayer and Monolayer Categories Results

As a precursor to the classification via machine learning algorithms, a PCA and a comparison between the mean spectra of monolayer and multilayer materials were conducted to determine whether discernible differences between the two material groups exist, which can be exploited for their differentiation into the classes monolayer and multilayer material.

The application of PCA onto the spectral information yielded three clear clusters. The evaluation of the PCA showed that the first principal components could explain approximately 80% of the variance. This result successfully classified multilayer, monolayer, and the sorter’s background into the three categories by machine learning algorithms. Figure 20 shows the result of this PCA. Here the three clusters can be seen. Green represents monolayer spectra, red represents multilayer spectra, and black represents the background material used in the trials. The monolayer and multilayer materials variance is described in dominant parts by the first principal component, further shown in the Pareto distribution diagram in the lower right corner. The first three principal components correspond with the spectral wavelengths of the separately examined spectra of 1038 nm, 1187 nm and 1309 nm that correspond to the second overtone of CH vibrations typical of CH_2_, CH_3_ and C=C chemical structures [27].

The evaluation of the spectral differences in the mean spectra taken from the monolayer and multilayer fraction yielded three spectral regions in which the mean spectra of monolayer and multilayer materials differ significantly. The comparison is visualised in Figure 21, which shows the mean multilayer spectrum in yellow, the mean monolayer spectrum in red and the three most pronounced differences. The first region where significant spectral differences can be seen is 1230 nm, corresponding to the second overtone of the CH bond [27]. Here the multilayer spectrum exhibits a more prominent peak than the monolayer fraction, possibly because of a different CH content within the two fractions. A similar difference can be observed at 1380–1410 nm, where the monolayer spectrum experiences a more pronounced dip than the multilayer fraction. This spectral region corresponds to the stretching and deformation vibrations of the CH bond of CH_2_ structures [27]. While these two differences expressed a similar characteristic, namely a dip or a peak, the third difference sees the two spectra deviating strongly from each other [27]. Between 1410 nm and 1440 nm, the multilayer spectrum exhibits a wave-like pattern while the monolayer spectrum rises until a peak is reached. This spectral region can be associated with vibrations of several chemical bonds as the first overtone of OH stretching vibrations, stretching and deformation vibrations of CH in CH_2_ and aromatic structures and the first overtone of NH vibrations. In particular, the shape of the spectra for the multilayer material would suggest that multiple peaks are present, and they might correspond to vibrations of aromatic or NH bonds typical of PET and PA, respectively. The spectra of the monolayer material would suggest possible vibration of one chemical bond type.

While the comparison of mean spectra of different materials cannot determine whether the differentiation of individual materials into the categories monolayer and multilayer is possible, it shows that differences between the two materials exist, which may be used to classify them accordingly.

After these preliminary examinations, the respective machine learning algorithms were used to classify the spectral data. Table 3 shows the success rate of each respective algorithm in correctly classifying the material into the classes multilayer, monolayer and background. All used algorithms show promising results apart from the k-means algorithm. This algorithm could not correctly identify the material, reaching an accuracy of only 60%. Amongst the others, the SVM and the SNN reached the highest accuracy. 

#### Prediction Speed

The NIR sorter can achieve a refresh rate of approximately 500 Hz, which can effectively be halved without substantial loss of information while recording 320 spatial pixels with a spectral resolution of 220 points. This recording speed means that approximately 80,000 spectra must be evaluated every second. A machine learning algorithm’s prediction speed is given as the number of observations processed per second, and its inverse would be the time taken for one prediction in seconds. The fastest examined machine learning algorithms were capable of prediction speeds of 83,000 observations per second, which would be fast enough to classify every pixel the spectral imaging camera recorded. It has to be noted that no pre-processing steps and additional computing time were considered for this calculation, reducing the number of spectra processable per second.

After evaluating the prediction speed and accuracy, a hierarchy of machine learning algorithms was established. Table 4 shows the percentage of correctly identified pixels and respective machine learning algorithm. With a PCA leaving three principal components for classification, the SVM outperformed the other algorithms regarding prediction speed and accuracy. This comparison is shown in Figure 22, which compares the examined algorithms and their success in classifying the test set. On the left, the prediction accuracy is presented. Here it can be seen that while all algorithms were able to label the spectra correctly in at least 80% of cases, the SVM, after PCA using the one versus one approach, could predict the material in 93% of cases correctly. Because these examinations aim to evaluate the algorithms for their applicability in a sorting operation, accuracy without prediction speed is irrelevant. Figure 22 shows on its right the comparison of the machine learning approaches concerning their time requirements per correct prediction. It can be seen that the introduction of a PCA prior to model generation decreased the time necessary to predict the label of a spectrum. Further, the PCA did not decrease accuracy. Therefore, the fastest algorithm was the SVM and the SNN with prior PCA using three principal components for prediction.

### 4.7. Visualisation of the Classification Results of the Shallow Neural Network

The comparison of the applied machine learning tools yielded two methods especially well suited to the classification of films. The SVM and the SNN were almost identical in prediction accuracy and speed when presented with unknown data. Though both methods are on equal footing on these metrics, the SNN is superior in terms of training time. The SVM took 260 s to train, while the SNN only took 16 s. While these specific times are highly dependent on the hardware used for training, the ratio between the training times is independent of the hardware used for training. It took almost 18 times longer to train the SVM. Due to this advantage of the SNN, it was used to classify film specimens. In the following, the classification results of the SNN are shown.

The following figures show the classification results of the films. Each pixel identified in the evaluated rectangle as monolayer is displayed in green, multilayer pixels are shown in red and pixels identified as background are black. 

Figure 23 shows the classification of a PE monolayer film. The SNN correctly identified most of the material. Areas with low spectral intensity were classified as background and are shown in black. A small number of pixels was wrongly classified as multilayer material. This issue is caused by the close resemblance of PE monolayer material´s spectra with PE/PP multilayer films, which can lead to misclassification.

Figure 24 illustrates the classification result of a PE/PET multilayer film. The specimen in question had an elongated form and some overexposure occurred, as shown by the bright sections in the image. The model had issues classifying the overexposed pixels, which can be seen by the red and green pixels, misclassified as mono- and multilayer film. Concerning classifying the specimen itself, the model was successful, shown by the resulting classification in red and the small number of misclassified pixels in green.

Figure 25 shows the classification result of a PE/PP multilayer packaging film. PE/PP multilayer materials challenge the classification model due to the close resemblance of the PE/PP multilayer spectrum and the corresponding monolayer spectra of PE and PP monolayer materials. The result is a rather large proportion of misclassified pixels, as shown in the figure. Despite these unfavourable circumstances, the model managed to classify most of the specimens correctly as multilayer material.

Figure 26 shows the unclassified specimen and the classification result of the multilayer cheese packaging. The neural network had issues with the low intensity of the recording in some areas, shown by the large proportion of pixels classified as background in black. The neural network correctly classified most of the specimen´s pixels where the intensity was sufficient for classification. Only a minuscule number of pixels were wrongly classified as monolayer pixels, shown by the green pixels in the classified image.

### 4.8. Application of FFT and Elimination of Frequencies

The application of FFT and subsequent elimination of interfering spectral abnormalities yielded improved spectra. These spectra regained their specific form used to categorise the respective materials. Figure 27 visualises the original spectrum before applying FFT and the following elimination of overlaying wave patterns. It can be seen that the spectrum exhibits almost no discernible patterns which could be used for classification. The characteristic peak at 1350 nm is insinuated but not pronounced. Contrarily, the characteristic peaks at 1150 nm and 1410 nm, expressed by the reference spectrum, are absent. After eliminating the overlying sine wave-like patterns, the fidelity of the spectrum to the reference spectrum improves. Although the peak expressed by the reference spectrum at 1150 nm could not be reproduced, the peak at 1350 nm becomes more pronounced and a second peak at 1410 nm becomes apparent. Further, the sine wave begins to form in the original spectrum at around 1390 nm and becomes less pronounced. 

The deviation from the reference spectrum could be reduced by up to 30%. This way, the information contained in PP spectra which were unuseable for the classification and generation of a separation model, could be extracted.

Finding the correct place for the filter has been automated, significantly reducing the workload for finding the correct filter placement. 

### 4.9. Spectral Library of Film Materials

During the creation of the machine learning tests and the sorting trials, an abundance of spectral information of film materials has been recorded. This spectral information has been stored in MAT-files. MAT-files are binary files that store workspace variables. This spectral library contains the spectral data of over 130 film specimens. These spectra and the necessary MATLAB code library to visualise the spectral images and to extract spectra from these files have been organised into a repository. This repository and the data therein may be used to create proprietary film sorting models for further trials. This spectral library expands the existing TrashNet-NIR library by adding film spectra. For access to the repository, the corresponding author may be contacted. 

## 5. Discussion

The sorting trials, the application of FFT and the machine learning approaches are discussed in the following. Further, the limitations of using a chute sorter to separate film specimens are evaluated and the possibility of incorporating the shown procedure in an integrated film separation process is elaborated upon.

### 5.1. Discussion of Sorting Trials

The sorting results indicate that the success rate of film sorting increases when reflecting backgrounds are used. The detection rate with a traditional non-reflecting glass chute did not reach 50%. With the introduction of a reflective chute, the detection rate reached over 70%, with better sorting results in every material category. This result supports the findings of previous experiments, which showed the improvement of film spectra using measurements in transflectance [2].

The increase in the detection rate is due to better useable spectra when the measurements are taken in transflectance mode. Furthermore, adding a reflective surface decreases the amount of radiation lost to transmission and enhances the spectral data quality available for classification and model creation. 

The transflection mode was only evaluated with a chute sorter. However, in material recovery facilities, belt sorters are usually used for their higher throughput and the continuous speed of the particles. The specimen´s speed depends on its density and shape on a chute sorter. While in this case the input material is film, it is not so much the particle density as the particle shape that is a problem. Films, in particular, are difficult to sort, as their low weight and large surface area make them prone to gliding, making their ballistics hard to predict and their ejection difficult. Though the improvement in spectral quality and sorting of films using the transflection mode could be shown, further evaluation of transflectance measurements with a belt sorter would be advisable.

In addition, some material classes have been underrepresented due to a lack of available specimens owing to low occurrence in the waste stream.

Finally, the created monolayer fraction could be further sorted into the respective monolayer materials, PE and PP. Out of this monomaterial feedstock, recyclate and subsequently test pieces for mechanical examinations could be produced. These tests, for example, tensile tests or Charpy tests, could then be used to assess the mechanical properties of the recyclate.

### 5.2. Discussion of the Application of Machine Learning Approaches

Implementing machine learning algorithms such as an SVM or a deep neural network showed great promise in classifying monolayer and multilayer materials. The prediction speed without a preliminary dimension reduction was insufficient to even theorise about their feasibility in an industrial setting. After implementing a dimension reduction using principal component analysis, the prediction speed increased substantially. In addition to an increase in prediction speed, prediction accuracy also saw an incremental increase. 

The correct classification of multilayer material without creating a specific model for each material class can be achieved by using common patterns among multilayer material. This is because machine learning methods can use these shared properties to detect multi-layered particles which can subsequently be ejected. Hence, machine learning is suited to be used for this purpose.

### 5.3. Discussion of the Application of FFT to Improve Spectra Overlain by Sine Wave Abnormalities

Because the implementation of reflective background materials only reduced the occurrence of sine wave noise, this issue still needed attention. The tedious search for the ideal cut-off point was replaced by a simple algorithm that finds the optimal position where the reconstructed spectrum comes closest to a reference spectrum.

The main problem with this approach is that it depends on knowledge of the polymer type of the material. Its purpose was to elaborate on the possibilities of using FFT to improve film spectra. Further research is needed to ensure that the system can improve a spectrum without prior knowledge of its polymer type by having generic reference spectra of polymers to compare the improved spectra against. It is not necessarily the case that the recreated spectrum needs to adhere to a spectrum of the same material class. Instead, the goal of the FFT process is to reduce or eliminate overlaying sine wave spectral noise. So, comparing the spectra with an adroitly chosen generic reference spectrum exhibiting no sine wave disturbances could be sufficient. The improved spectrum could then be used for the actual classifying process. Further, only the application of a low pass filter has been described in this article as it extraordinarily improved the spectral quality. Additional trials may show that a supplementary implementation of a high pass filter may improve the spectral quality further though this has not been evaluated.

### 5.4. Discussion of an Integrated Process

Combining all processes shown in this work may be used to classify film spectra. First, the spectral image is taken in transflection and evaluated. The spectra used for classification are then classified either as suitable or unsuitable. If a spectrum is unsuitable for further classification due to sine wave noise caused by destructive interferences, the spectrum is improved via the shown FFT. Based on the spectra, the material is then classified by an SVM or neural net. Depending on the classification result, monolayer or multilayer film, the material is subsequently handled accordingly. In the case of monolayer material, further classification into the respective material groups via NIR is undertaken to create a monomaterial input stream for supplementary recycling processes. Two options are available for discussion if the material is classified as a multilayer. The material can either be thermally utilised or used as a feedstock for chemical recycling. Figure 28 shows a flowchart for this method.

## 6. Conclusions

NIR sorting success depends on the availability of high-quality spectral information. Traditional approaches struggle to provide spectra with high fidelity, as shown in the sorting trials lacking reflective backgrounds. Introducing reflecting backgrounds enables measurements in transflection, permitting the separation of monolayer and multilayer materials. This approach yielded an increase in detection rate from 46% to 74% with an aluminium reflector. Implementation of a copper reflector improved the detection rate further to 78%. Apart from an increase in the average detection rate, the recognition of every individual material increased with the introduction of reflective backgrounds. These findings support existing results that by increasing the reflectivity of the background material and the coinciding measurements in transflection, the sorting success of 2D materials can be increased.

Existing findings regarding the application of FFT to improve the spectral quality further were deepened. We proposed a method to apply FFT to spectra in order to eliminate destructive interference which in turn reduces the (manual) time demand. The improved spectra can then be used in machine learning methods to separate monolayer from multilayer materials. This adoption of machine learning methods was performed after the applied PCA showed characteristic differences between the spectra of mono- and multilayer films, regardless of their material composition. These overarching differences were used to train machine learning models. The trained machine learning models could correctly categorise mono and multilayer materials without the need to include every combination of multilayer materials in the training set. The computation times were low enough to consider the applicability of these methods for inline classification. Here, additional research is needed with more potent hardware.

## Figures and Tables

**Figure 1 polymers-14-03926-f001:**
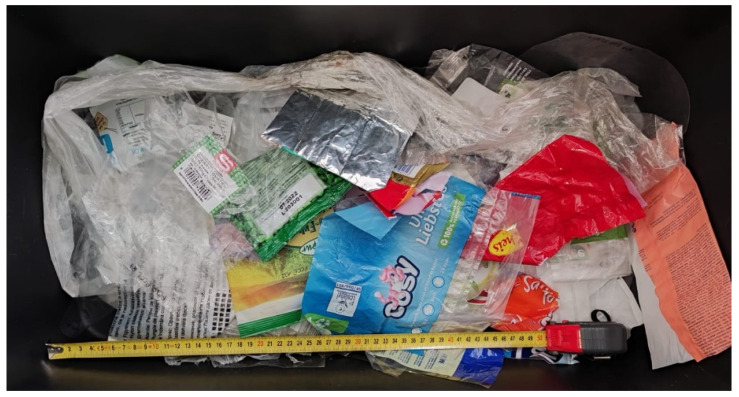
Fraction of small films waste.

**Figure 2 polymers-14-03926-f002:**
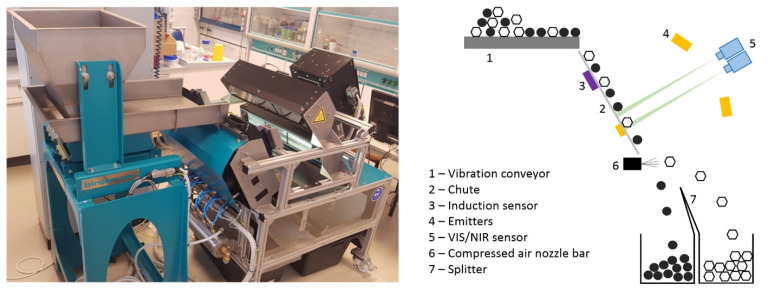
Experimental sensor-based sorting setup with the use of near-infrared spectroscopy.

**Figure 3 polymers-14-03926-f003:**
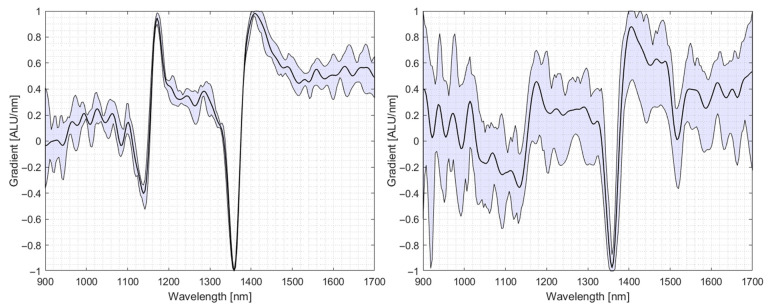
Comparison of spectral variability and characteristic peaks of a PE film when measured in transflection (**left**) and measured in reflectance (**right**).

**Figure 4 polymers-14-03926-f004:**
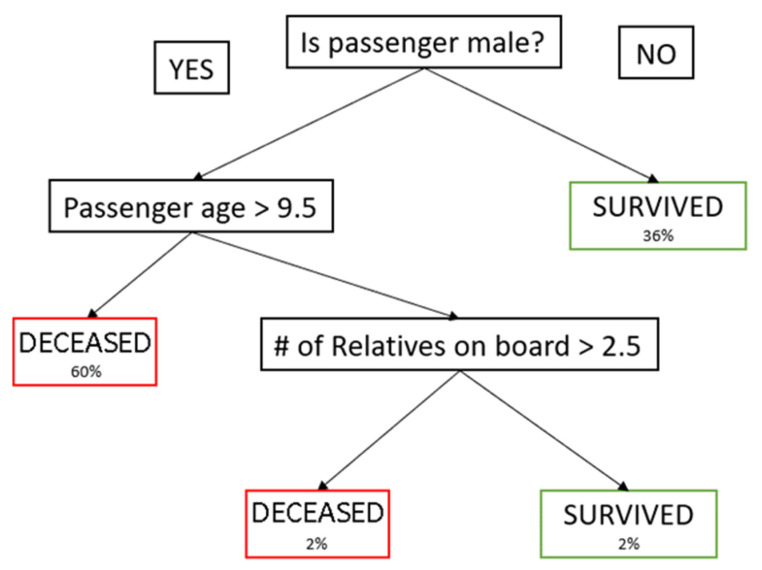
Example of a decision tree.

**Figure 5 polymers-14-03926-f005:**
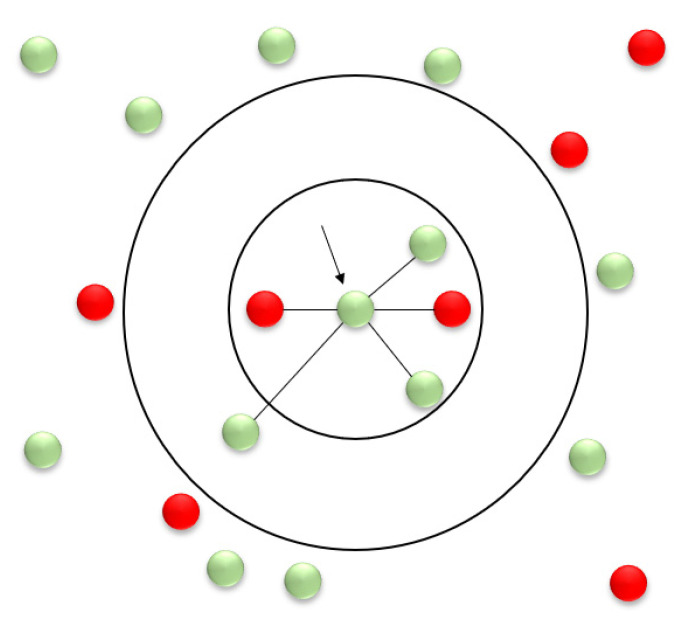
Example for k-nearest neighbour classification using k = 5 neighbours.

**Figure 6 polymers-14-03926-f006:**
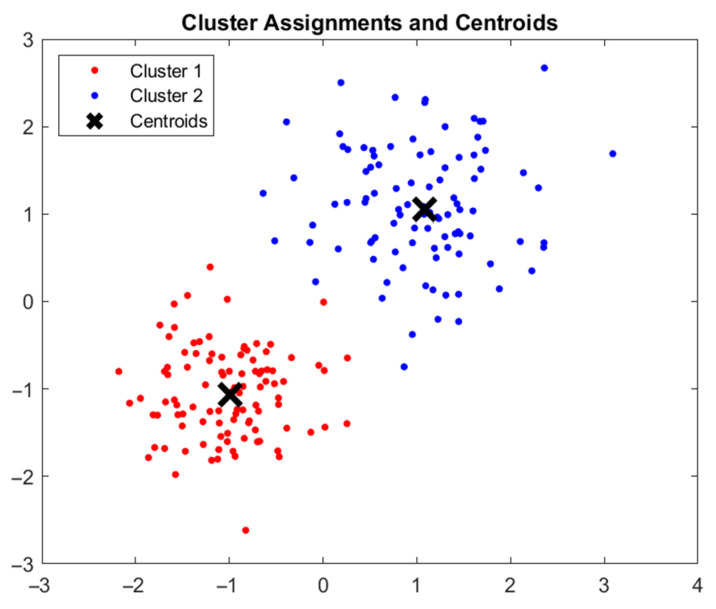
Example of a k-clusters clustering problem.

**Figure 7 polymers-14-03926-f007:**
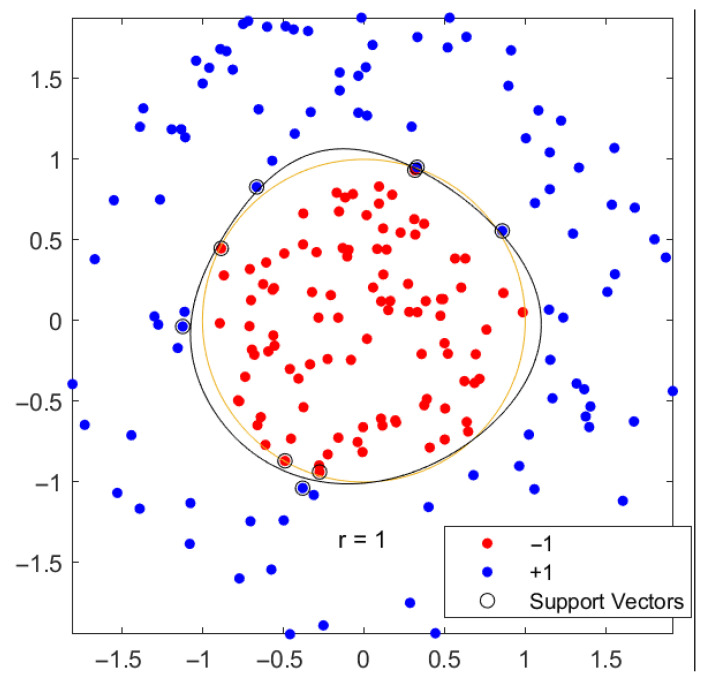
SVM classification using a nonlinear hyperplane and the classification result and the used support vectors to create the hyperplane.

**Figure 8 polymers-14-03926-f008:**
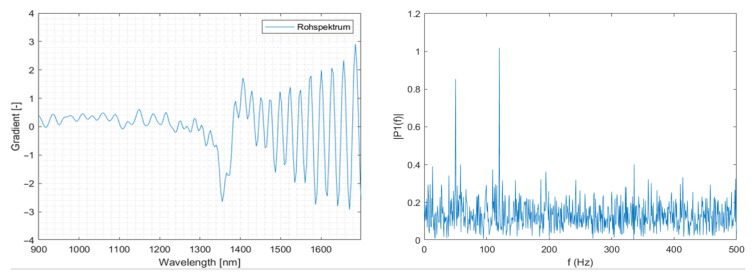
Spectral and Fourier depiction of a noisy signal.

**Figure 9 polymers-14-03926-f009:**
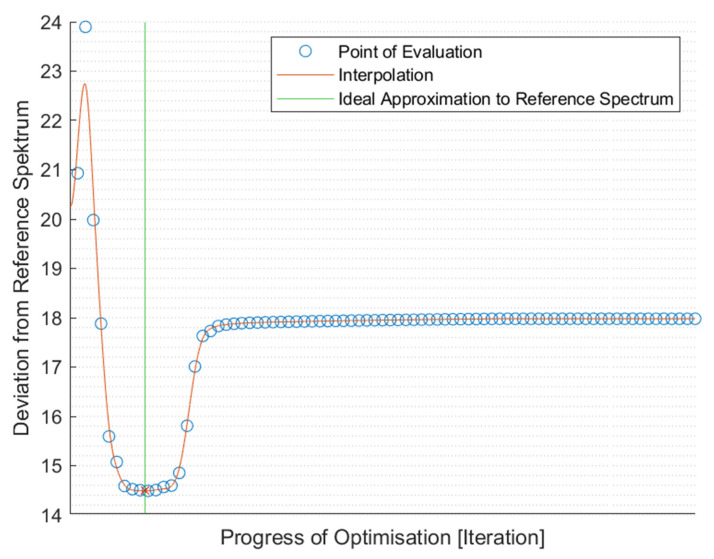
Progress of the optimisation over the deviation of the resulting spectrum from the reference spectrum.

**Figure 10 polymers-14-03926-f010:**
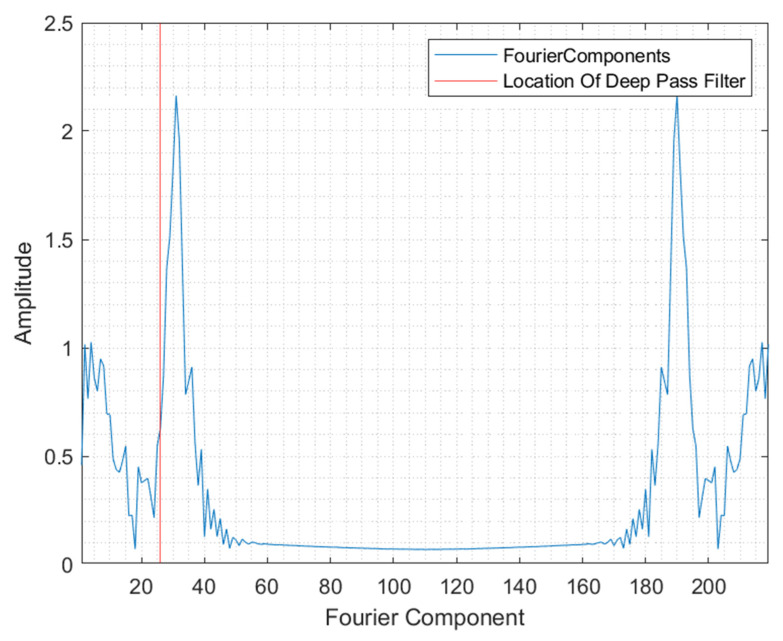
Fourier representation of the original spectrum.

**Figure 11 polymers-14-03926-f011:**
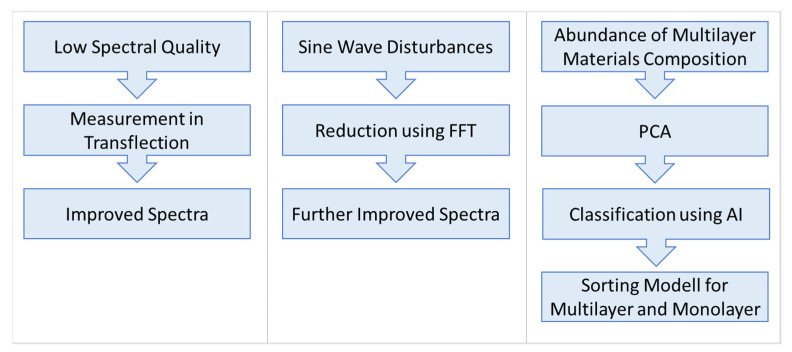
Summary of encountered problems when sorting films and the applied solutions.

**Figure 12 polymers-14-03926-f012:**
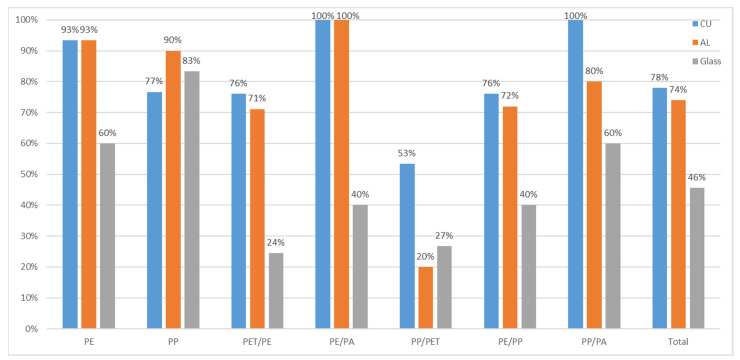
Detection rate with different reflectors concerning different materials.

**Figure 13 polymers-14-03926-f013:**
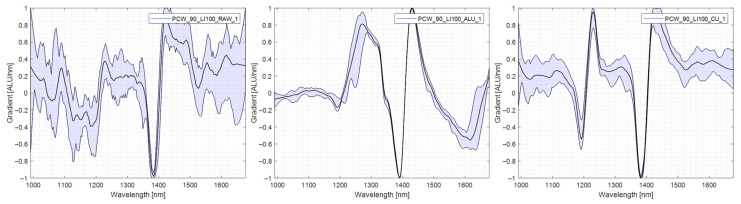
Comparison PE spectra, **left**: no reflector, middle: aluminium reflector, **right**: copper reflector.

**Figure 14 polymers-14-03926-f014:**
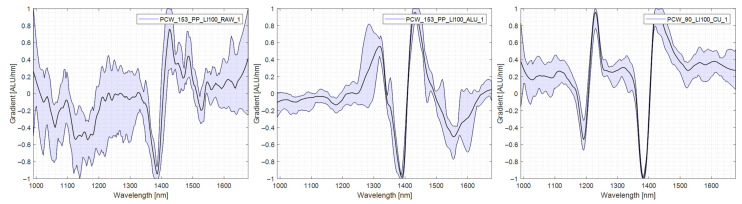
Comparison of PP spectra, **left**: no reflector, middle: aluminium reflector, **right**: copper reflector.

**Figure 15 polymers-14-03926-f015:**
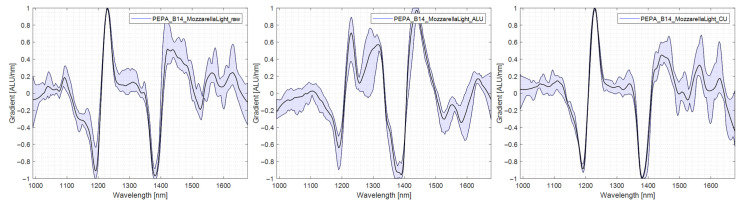
Comparison of PE/PA spectra, **left**: no reflector, middle: aluminium reflector, **right**: copper reflector.

**Figure 16 polymers-14-03926-f016:**
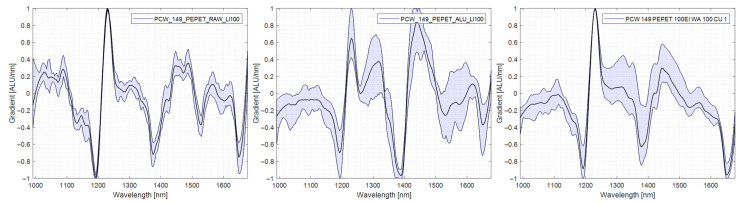
Comparison PE/PET spectra, left: no reflector, middle: aluminium reflector, right: copper reflector.

**Figure 17 polymers-14-03926-f017:**
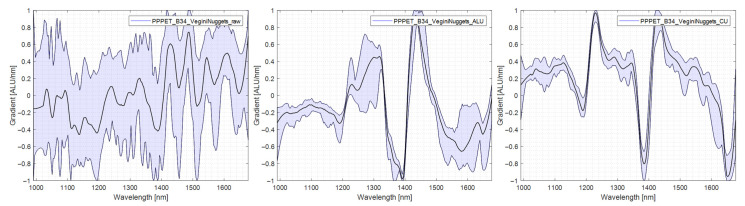
Comparison PP/PET spectra, left: no reflector, middle: aluminium reflector, right: copper reflector.

**Figure 18 polymers-14-03926-f018:**
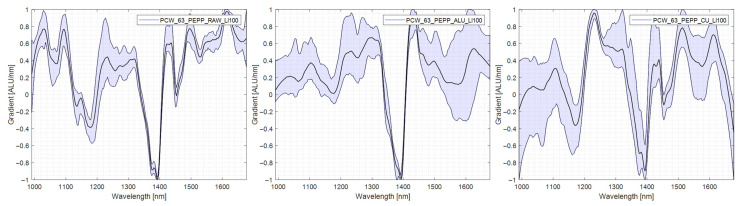
Comparison PE/PP spectra, left: no reflector, middle: aluminium reflector, right: copper Reflector.

**Figure 19 polymers-14-03926-f019:**
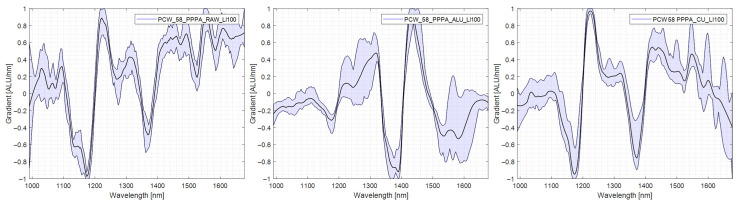
Comparison PP/PA spectra, left: no reflector, middle: aluminium reflector, right: copper reflector.

**Figure 20 polymers-14-03926-f020:**
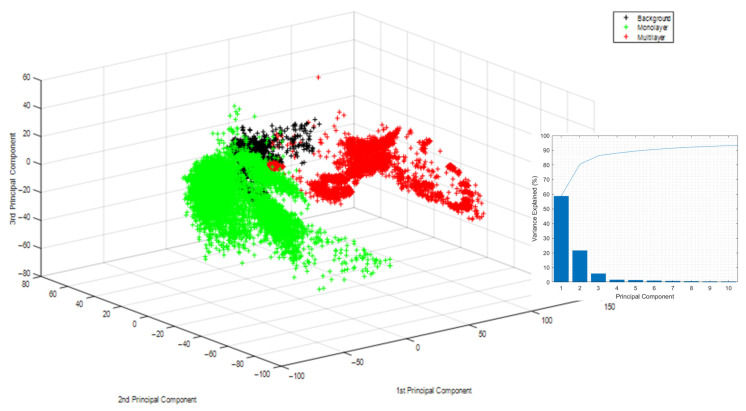
Result of the principal component analysis of approximately 17,000 spectra of monolayer, multilayer and background material to discern their sortability.

**Figure 21 polymers-14-03926-f021:**
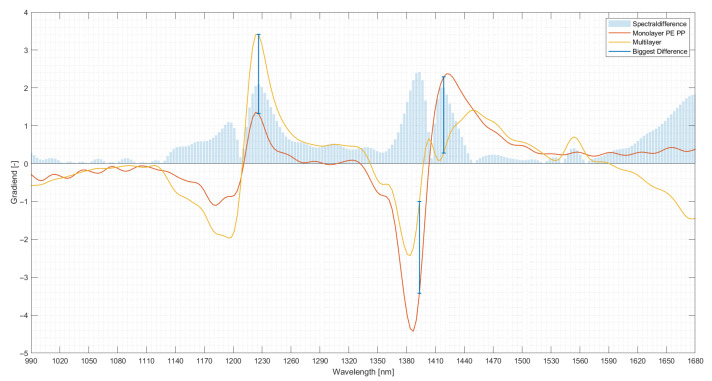
Differences in the mean spectra of multilayer and monolayer materials.

**Figure 22 polymers-14-03926-f022:**
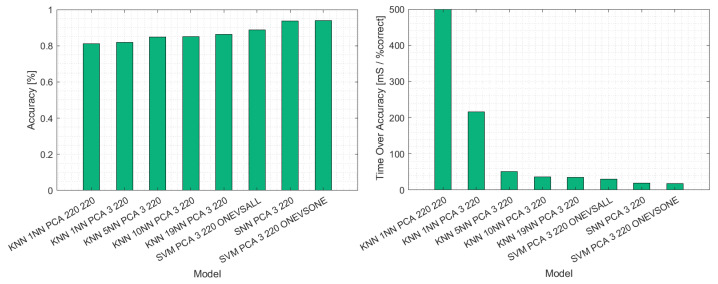
Comparison of the different machine learning algorithms used for classifying monolayer and multilayer materials in the test set.

**Figure 23 polymers-14-03926-f023:**
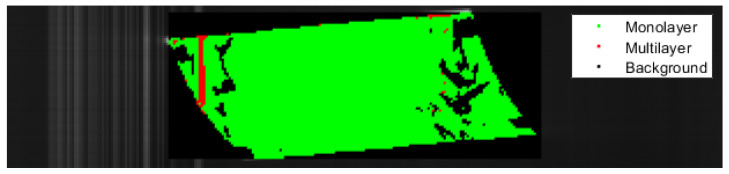
Classification of a PE monolayer film with the SNN.

**Figure 24 polymers-14-03926-f024:**
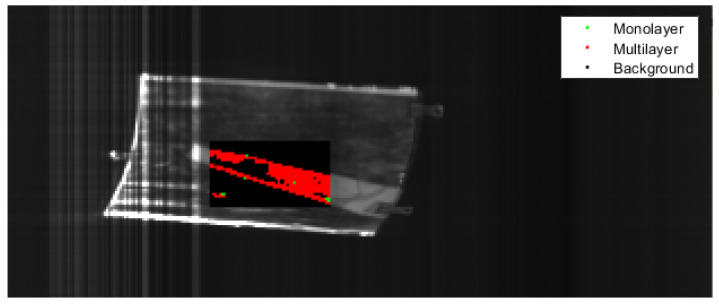
Classification of PE/PET multilayer film with the SNN.

**Figure 25 polymers-14-03926-f025:**
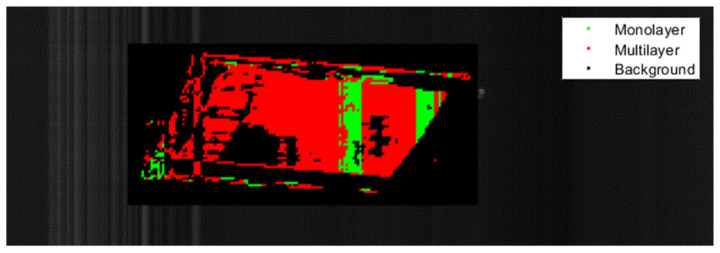
Classification of PE/PP multilayer film with the SNN.

**Figure 26 polymers-14-03926-f026:**
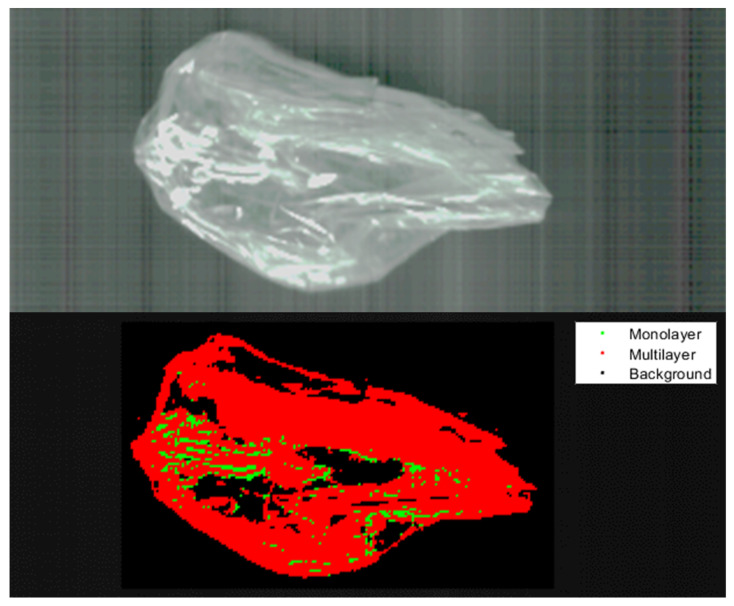
Classification of a cheese packaging film not used in training and testing with the SNN.

**Figure 27 polymers-14-03926-f027:**
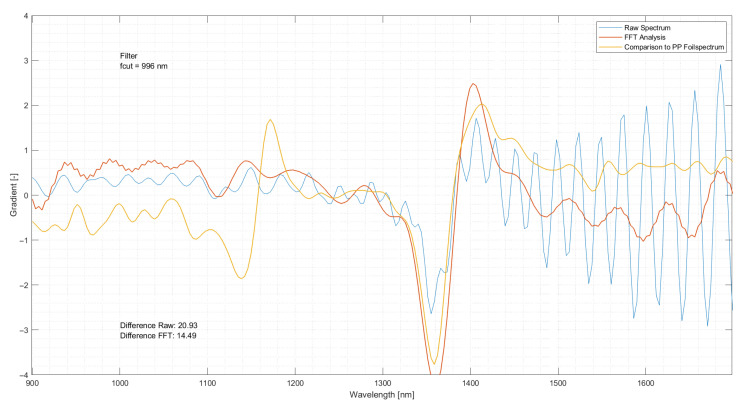
Difference between the raw spectrum (blue) and an improved spectrum (red) and their comparison to a reference spectrum (yellow).

**Figure 28 polymers-14-03926-f028:**
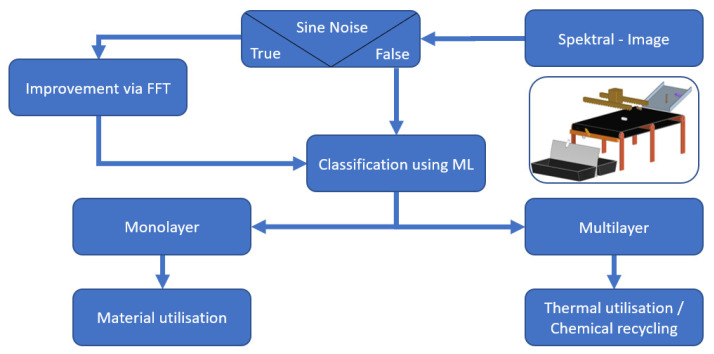
Integrated film recycling process.

**Table 1 polymers-14-03926-t001:** List of mono- and multilayer materials used in the sorting trials.

Materials	Recycling Label	Share
Polyethene	PE	9 wt.%
Polypropylene	PP	31 wt.%
Polyethene + polyethylene terephthalate	PE/PET	28 wt.%
Polyethylene + polyamide	PE/PA	6 wt.%
Polyethylene + polypropylene	PE/PP	16 wt.%
Polypropylene + polyethylene terephthalate	PP/PET	9 wt.%
Polypropylene + polyamide	PP/PA	1 wt.%

**Table 2 polymers-14-03926-t002:** List of pre-processing and spatial-processing.

Pre-Processing	Spectral-Processing
Bad pixel replacement	Calculation of the first derivative
Intensity calibration	Smoothing
Noise suppression	Normalisation
Spatial correction	

**Table 3 polymers-14-03926-t003:** Ejection rates with different reflectors.

Film Material	Ejection Rate Copper Reflector [%]	Ejection Rate Aluminium [%]	Ejection Rate No Reflector [%]	Average Ejection Rate [%]
PE	93	93	60	82
PP	77	90	83	83
PET/PE	76	71	24	57
PE/PA	100	100	40	80
PP/PET	53	20	27	33
PE/PP	76	72	40	63
PP/PA	100	80	60	80
Total	78	74	46	66

**Table 4 polymers-14-03926-t004:** Correctly identified pixels and respective machine learning algorithm.

Algorithm	Correctly Identified Pixels
Decision tree	98.15%
k-nearest neighbour	98.17%
Neural net	99.47%
Support vector machine	99.63%
**Shallow neural network**	**99.90%**
k-means	~60%

## Data Availability

The data presented in this study are available on request from the corresponding author.

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
