# Peer review of "Evaluation of Improvements in the Separation of Monolayer and Multilayer Films via Measurements in Transflection and Application of Machine Learning Approaches"

_polymers, 2022, doi:10.3390/polym14193926_

Round 1

Reviewer 1 Report

The manuscript submitted by the authors is based on an experimental setup very recently proposed by the authors (Waste Management,144, 543–551, 2022). In the specific application, the authors pointed out the difficulty in identifying multi-layered plastic films in view of their recycling and proposed a procedure for sorting this kind of material from single-layer films.

The manuscript is appreciable for the attention given by the authors to the ‘practical’ aspects of the analysis, in particular the need for a rapid classification of the materials based on their FTIR spectra which makes the use of chemometric tools unavoidable. On the other hand, the description of these methods is redundant on one hand (the authors use common algorithms implemented in preset Matlab packages, there is no need for presenting out-of-context examples as the Titanic data set) but, on the other hand, in some case also difficult to justify. I’m referring in particular to the use of deep-learning algorithms (that have nothing to do with the representation of human brain functionality!) when a shallow artificial neural network would have been more appropriate, reducing the risk of overfitting and providing results not critically depending on the network architecture (“The neural net can be regarded as an outlier because its success rate dramatically depends not only on the training sets available but also on the neural net architecture.”). It is strange to see that, after this premise, most of the examples reported (fig. 24-26) were referring to the artificial neural network results.

The discussion on the classification methods should be, in my opinion, completely rewritten. The methods used are well known and only the results of the most performing algorithm (which very probably will be the simple backpropagation shallow neural network) should be commented.

While revising the manuscript, the authors should also check the following issues:

-       Figure 3 is essentially the same as published in the 2022 paper on Waste Management

-       The Author Contributions section was not completed

The English should be improved.

Reviewer 2 Report

The authors have improved a detection rate of 78%. Authors can compare this detection rate with other detection methods, to establish the detection quality.

Authors can explain if are working with other reflectors.

Round 2

Reviewer 1 Report

I'm very glad to see that my suggestions were useful for the authors. The manuscript can be published in present form without further modifications.